# Particle Size Imbalance Index from Compositional Analysis to Evaluate Cereal Sustainability for Arid Soils in Eastern Algeria

**Siham Zaaboubi [1,*], Lotfi Khiari [2,3], Salah Abdesselam [1], Jacques Gallichand [3], Fassil Kebede [2] and Ghouati Kerrache [1]**

[1]    LAPAPEZA Laboratory, Department of Agriculture, Institute of Veterinary and Agricultural Sciences, Batna 1 University, aisles May 19, Route de Biskra, Batna 05000, Algeria; aksalah2001@yahoo.fr (S.A.); kerrache_g20@yahoo.fr (G.K.)

[2]    Soil and Fertilizer Research in Africa, Mohammed VI Polytechnic University, Lot 660, Hay Moulay Rachid, Ben Guerir 43150, Morocco; lotfi.khiari@um6p.ma (L.K.); Fassil.KEBEDE@um6p.ma (F.K.)

[3]    Department of Soils and Food Engineering, Faculty of Agriculture and Food Sciences, Paul-Comtois Building. 2425, Rue de l'Agriculture, Laval University, Quebec, QC G1A 0A6, Canada; lotfi.khiari@fsaa.ulaval.ca (L.K.); jacques.gallichand@um6p.ma (J.G.)

*    Correspondence: siham.zaaboubi@univ-batna.dz; Tel.: +213-658311935

**Abstract:** For homogeneous fertilization and crop management practices, this work hypothesized that texture could influence cereal yield, particularly in dry regions. Particle size analysis could help improve knowledge of the soil-plant relationship to obtain favorable conditions for better yield. The objective of this work is to develop a single granulometric index for durum wheat (*Triticum durum*) that is well correlated with yield. For this purpose, 350 independent samples of cereal soils from eastern Algeria were taken and the recorded yields were linked to these samples. The cutoff yield, which separates sub-populations with acceptable yield from those with less acceptable yield, was determined from the inflection point of the cumulative variance ratio functions related to yield by the Richards' equation. The result obtained is 2.0 Mg.ha$^{-1}$, with a theoretical critical chi-square value of 4.2, close to 4.6, which is the critical value of $r^2_{granulo}$ as obtained by the Cate-Nelson procedure. The five-granulometric indices were found to be symmetrical around zero as follows: ±0.83 for clay ($I_C$), ±1.73 for fine silt ($I_{FL}$), ±0.31 for coarse silt ($I_{CL}$), ±0.44 for fine sand ($I_{FS}$), and ±1.30 for coarse sand ($I_{CS}$). The two fractions that most influence the textural imbalance are fine silt ($I_{FL}$) and coarse sand ($I_{CS}$), with a contribution of 41% and 37%, respectively. The critical single imbalance index $r^2_{granulo}$ can be used for determining cereal suitability for soils in the arid region of eastern Algeria. The lower the $r^2_{granulo}$ is, the better the soil for cereal crops.

**Keywords:** soil texture; compositional analysis; durum wheat; eastern Algeria

## 1. Introduction

According to statistics [1], cereal production in Algeria has grown from 8 to 51 million quintals in half a century. This resulted in intensive tillage and an in the introduction of fallow years [2], exposing the soils to compaction and to degradation of surface horizons [3–5]. This intensification, therefore, prompted particular attention to the quality and overall health of cereal soils. Non-irrigated cereal-growing regions of eastern Algeria are characterized by the uniformity of crop management techniques with two-year rotations (biennial crop rotation), following the cereal crop and fallow. They are also characterized by the uniform application of mineral fertilizers, i.e., N-P$_2$O$_5$-K$_2$O of 30-69-0 at seeding in the form of DAP (di-ammonium phosphate), 30-0-0 at

tillering, and 30-0-0 at heading in the form of ammonium nitrate or other simple nitrogen fertilizer [6]. Being specific to this region and these practices, the fertility parameter that seems to influence cereal yields is rather physical, such as soil texture. Soil texture contributes to the inherent quality of the soil, and is virtually unchangeable by soil management. To ensure better monitoring of the sustainability of soil management practices, De la Rosa and Sobral [7] proposed the quantitative measurement of soil properties. Among these, soil texture is at the center of the physical characteristics on which several other important soil properties depend [8]. According to several authors [9–11], the soil texture controls many important ecological, hydrological, and geomorphic processes such as water retention, ion exchange, nutrient uptake, and crop yield. Thus, the productivity of soils and the adaptation of crops to the textural composition of soils are some of the best solutions [12], especially in conditions where water is limiting. In fact, Wambeke [13] pointed out that sandy soils are much less productive than other textures. Also, He et al. [14] reported that harvests for clay soil are about 20% higher than those for loamy soils. Paradoxically, Katerji and Mastrorilli [15] have shown opposite trends, with higher yields in loamy soils than in clay soils. For some crops, no significant differences in yield are observed for the two soils. Moreover, Xu et al. [16] recommended carefully examining the textural distribution for the choice of crops. The particle size analysis comprises several fractions that lend themselves well to compositional analysis. According to Parent et al. [17], this analysis avoids three statistical inefficiencies: redundancy of information by calculating the difference between a component and the sum of the other components [18], sub-compositional inconsistency, and non-normal data distribution [19]. It is, therefore, hypothesized that the compositional transformation of the five particle sizes (clay, fine loam, coarse loam, fine sand, and coarse sand) into a global index, noted $r^2_{granulo}$ [20–22], is relevant if this global index can be related to cereal yields. This $r^2_{granulo}$ index could then help the scoring process to assess soil health and cereal suitability.

The objective of this study is to amalgamate the five-particle size components into a single index $r_{granulo}^2$ better suited to identifying the zones of better cereal productivity in eastern Algeria.

## 2. Description of the Study Area

As shown in Figure 1, the study area is located in eastern Algeria, limited by 5°12′1.72″ and 7°19′83″ East longitude and 35°21′27.65″and 36°11′55.09″ North latitude.

The climate is characterized by a low rainfall of high spatial and temporal heterogeneity, which is a characteristic of the Mediterranean climate.

The soils of the study area are mainly rich in carbonates (calcareous soils) with an alkaline pH found on alluvial or sedimentary layers. They are classified among calcids and cambids according to the Soil Taxonomy (2010). This USDA classification defines Cambides as poorly developed aridisols with a shallow Cambic horizon, and Calcids as aridisols that have inherited calcium carbonate from the parent rock or by addition as dust, or both. Most of those soils are used for dry cultivation of cereals. The cereal-fallow system occupies a large portion the farming systems. Wheat and barley are the dominant crops.

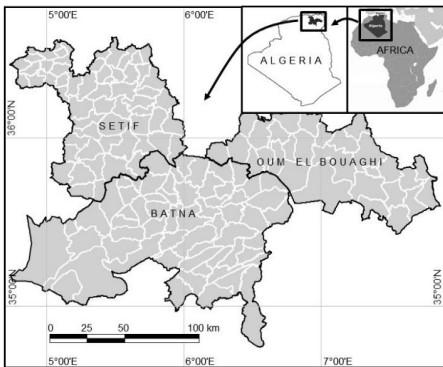

**Figure 1.** Location of studied areas.

## 3. Sampling and Soil Analysis

Soil samples were taken from 350 plots in autumn, over two agricultural seasons 2013/2014 and 2014/2015. The determination of the sampling units in each plot was carried out randomly according to the Pauwels et al. [23] method. These samples were taken before plowing to a depth of 30 cm (tilled horizon). These soil samples were dried in open air, crushed manually, and sieved through a 2 mm sieve. Soil properties (Table 1) have been determined by various methods. The particle sizes (clay, fine loam, coarse loam, fine sand, and coarse sand) were determined by the sedimentation method [24] after the removal of carbonate and organic substances.

**Table 1.** Descriptive statistics of soil properties of the study area.

|  | Mean | Median | Range | Standard Deviation | Variance Coefficient (%) |
|---|---|---|---|---|---|
| C ($g100g^{-1}$) | 24.6 | 24.0 | 2.1–52.8 | 11.6 | 47.2 |
| FL ($g100g^{-1}$) | 22.2 | 21.8 | 0.5–54.8 | 10.7 | 48.1 |
| CL ($g100g^{-1}$) | 17.0 | 17.0 | 5.1–37.1 | 6.7 | 39.6 |
| FS ($g100g^{-1}$) | 26.0 | 26.4 | 0.02–64.7 | 13.6 | 52.3 |
| CS ($g\ 100g^{-1}$) | 10.2 | 11.2 | 0.1–55.3 | 8.0 | 78.9 |
| BD ($gcm^{-3}$) | 1.2 | 1.1 | 1.00–1.8 | 0.1 | 15.9 |
| P ($g100g^{-1}$) | 52.7 | 54.7 | 30–61.5 | 7.5 | 14.3 |
| EC ($dS\ m^{-1}$) | 0.4 | 0.2 | 0.01–3.9 | 0.6 | 149.5 |
| $pH_{water}$ | 8.2 | 8.2 | 7–8.9 | 0.3 | 3.7 |
| $(CaCO_3)_{Total}$ ($g100g^{-1}$) | 28.8 | 26.4 | 0–65.6 | 15.7 | 54.6 |
| $(CaCO_3)_{active}$ ($g\ 100g^{-1}$) | 14.3 | 15.0 | 0–36.0 | 8.6 | 60.1 |
| $OC_{Total}$ ($g100g^{-1}$) | 1.41 | 1.3 | 0.43–2.8 | 0.49 | 35.04 |
| $N_{Total}$ ($g100g^{-1}$) | 0.1 | 0.1 | 0.04–0.3 | 0.0 | 39.4 |
| C/N | 11.1 | 9.4 | 4.9–30.9 | 5.2 | 47.0 |
| $P_{Olson}$ ($mg.\ kg^{-1}$) | 23.2 | 9.4 | 1.0–138.0 | 118.3 | 509.8 |
| $Ca_{Exch}$ ($cmol.\ kg^{-1}$) | 39.0 | 38.2 | 32.4–48.8 | 3.6 | 9.4 |
| $Mg_{Exch}$ ($cmol\ kg^{-1}$) | 1.9 | 1.9 | 0.05–5.6 | 1.2 | 63.8 |
| $Na_{Exch}$ ($cmol.\ kg^{-1}$) | 0.4 | 0.4 | 0.01–4.0 | 0.5 | 108.3 |
| $K_{Exch}$ ($cmol.\ kg^{-1}$) | 7.7 | 1.3 | 0.1–48.7 | 15.8 | 204.0 |
| CEC ($cmol.\ kg^{-1}$) | 49.2 | 42.5 | 34.0–98.4 | 17.5 | 35.6 |

C: Clay; FL: Fine Loam; CL: Coarse Loam; L: Loam; FS: Fine sand; CS: Coarse Sand; S: Sand; BD: Bulk density; P: Porosity; EC: Electrical Conductivity; $(CaCO_3)_{Total}$: Total $CaCO_3$; $(CaCO_3)_{active}$: Active $CaCO_3$; $OC_{Total}$: Total Organic Carbon; $N_{Total}$: Total Nitrogen; $P_{Olson}$: available Phosphorus; Exch: exchangeable; CEC: Cation Exchange Capacity.

After the particle classes were separated by sedimentation, a sample was taken by a Robinson pipette for the fine fractions <50 μm (clay, loam, and coarse loam) and by wet sieving through two sieves (50 and 200 μm) to determine the fractions of larger size (fine and coarse sands). The apparent volumetric mass was measured by the cylinder method [24]. The 1 M ammonium acetate method at pH 7 [25] was used to determine the contents of exchangeable bases (Ca, Mg, Na, and K). The total organic carbon was determined according to the modified Walkley and Black method [26]. The total nitrogen determination was carried out by the Kjeldahl method [26]. The available phosphorus was determined by the Olsen method [26], soil pH was measured with a soil-to-water ratio of 1:1 [25], and the total limestone determination was carried out by the calcimeter method [26].

## 4. Statistical Analysis

The cutoff values for distinguishing between high and low yields were determined by the cumulative variance ratio functions $F_i^C(V_X)$, developed by Khiari, Parent, and Tremblay [20]. The sigmoid adjustment, translating the evolution of these $F_i^C(V_X)$ as a function of the yields, was done using the five parameters Richards' equation (Table 2). This equation is very flexible, but was only used to provide an empirical adjustment to infer the inflection point, which defines the change in concavity. This inflection point is the yield cutoff used to divide the productive sub-population from the less productive sub-population [20].

**Table 2.** Generalized logistic function or Richards' curve.

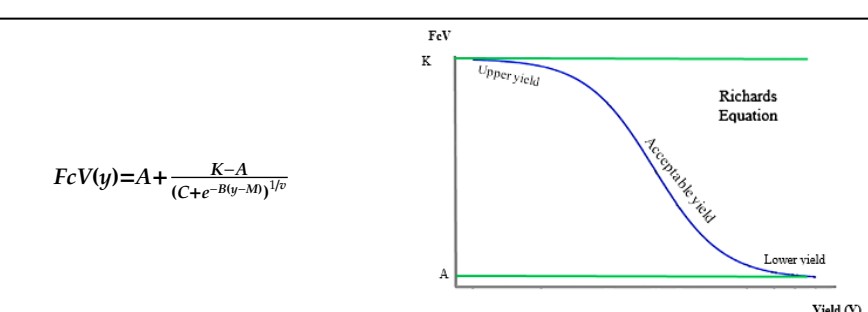

$$FcV(y) = A + \frac{K-A}{(C + e^{-B(y-M)})^{1/v}}$$

| $F_i{}^c(V_X)$ | A | K | M | B | V | $F_i{}^c(V_X)$-IP | Yield-IP |
|---|---|---|---|---|---|---|---|
| $F_i{}^c(V_C)$ | 102 | 5.54 | 4.66 | 0.134 | 0.229 | 63.0 | 1.56 |
| $F_i{}^c(V_{FL})$ | 102 | 1.21 | 5.19 | 0.134 | 0.224 | 61.3 | 1.64 |
| $F_i{}^c(V_{CL})$ | 113 | 0.0702 | 2.51 | 0.0994 | 0.202 | 67.5 | 1.36 |
| $F_i{}^c(V_{FS})$ | 113 | −2.81 | 5.05 | 0.0817 | 0.296 | 64.9 | **1.99** |
| $F_i{}^c(V_{CS})$ | 102 | 4.86 | 4.18 | 0.122 | 0.218 | 62.6 | 1.66 |

$F_i{}^c(V_X)$: the cumulative variance ratio function (%) for the component X; A: the lower asymptote; K: the upper asymptote; M: mean; B: the growth rate = 3 (Lower yield, Acceptable yield, and Upper yield); C: typically takes a value of 1; V: affects near where asymptote maximum growth occurs >0; $F_i{}^c(V_X)$-IP: The critical value of the cumulative variance ratio function (%) for the X component at the inflection point (IP); Yield-PI: critical yield in Mg ha$^{-1}$ at the inflection point (IP); $F_i{}^c(V_C)$: cumulative variance ratio function for Clay; $F_i{}^c(V_{FL})$: cumulative variance ratio function for fine loam; $F_i{}^c(V_{CL})$: cumulative variance ratio function for coarse loam; $F_i{}^c(V_{FS})$: cumulative variance ratio function for fine sand; and $F_i{}^c(V_{CS})$: cumulative variance ratio function for coarse sand. The value in bold was chosen to discriminate between the productive and the less productive population.

The granulometric indices were calculated using the Excel package (Microsoft, 2013) according to the theory presented below (Equations (1)–(6)).

The critical thresholds for these indices were obtained using the Cate-Nelson binary partition [26]. By a simple iterative process, we obtain a series of values of the sums of the squares for the multitudes of divisions performed at different levels of the independent variable X. The critical level of X is that where the sum of squares is the maximum. For the dependent variable Y, the critical level of Y is that where the set of points in the error quadrants (false positive and false negative) is the minimum. The calculations of these binary partitions were carried out with R software (R 3.6.2) [27].

## 5. Rationale

The granulometric property of the soil forms a texture simplex ($S^5$), since it is formed of five components whose sum is equal to 100% according to the theory of compositional analysis [28]:

$$S^5 = \{(C, FL, CL, FS, CS): C > 0, FL > 0, CL > 0, FS > 0, CS > 0; C + FL + CL + FS + CS = 100\} \quad (1)$$

where C = clay, FL = fine loam, CL = coarse loam, FS = fine sand, and CS = coarse sand are particle size proportions all expressed as a percentage.

The particle size fractions become invariant on the scale when divided by their geometric mean (G) defined as follows (Equation (2)):

$$G = [C \times FL \times CL \times FS \times CS]^{1/5}. \quad (2)$$

$V_C$, $V_{FL}$, $V_{CL}$, $V_{FS}$, and $V_{CS}$ are the log-centered ratios of these five particle size components on their geometric means, respectively (Equation (3)).

$$V_C = \left(\frac{C}{G}\right), V_{FL} = \left(\frac{FL}{G}\right), V_{CL} = \left(\frac{CL}{G}\right), V_{FS} = \left(\frac{FS}{G}\right), V_{CS} = \left(\frac{CS}{G}\right) \quad (3)$$

$$V_C + V_{FL} + V_{CL} + V_{FS} + V_{CS} = 0 \qquad (4)$$

By definition, the sum of the five particle size components is equal to 100% (Equation (1)) and the sum of the log-centered ratios must be zero (Equation (4)).

The first texture norms are the means ($V^*_C$, $V^*_{FL}$, $V^*_{CL}$, $V^*_{FS}$, and $V^*_{CS}$) and standard deviations ($SD_C$, $SD_{FL}$, $SD_{CL}$, $SD_{FS}$, and $SD_{CS}$) of the log-centered ratios of the five particle size components.

The five log-centered ratios are normalized as follows:

$$I_C = \frac{\left(V_C - V^*_C\right)}{SD_C}, \ I_{FL} = \frac{\left(V_{FL} - V^*_{FL}\right)}{SD_{FL}}, \ I_{CL} = \frac{\left(V_{CL} - V^*_{CL}\right)}{SD_{CL}}, \ I_{FS} = \frac{\left(V_{FS} - V^*_{FS}\right)}{SD_{FS}}, \ I_{CS} = \frac{\left(V_{CS} - V^*_{CS}\right)}{SD_{CS}} \quad (5)$$

where $I_c$, $I_{FL}$, $I_{CL}$, $I_{FS}$, and $I_{CS}$ are the particle size indices.

The particle size indices as defined by the Equation (5) are standardized and linearized variables in a five dimensional space [28].

The single particle size imbalance index of a diagnosed sample is the $r^2_{granulo}$ calculated according to the following equation:

$$r^2_{granulo} = I^2_C + I^2_{FL} + I^2_{Cl} + I^2_{FS} + I^2_{CS}. \qquad (6)$$

The square sum of these five standardized and independent particle size indices provides a new variable that follows the chi-square distribution law with 5 degrees of freedom [29].

According to the previous Equations (5) and (6), the closer the particle size indices are to zero (the values calculated by $r^2_{granulo}$ and the chi-square), the higher the probability of obtaining a good yield.

## 6. Results and Discussion

### 6.1. Step 1. Selection of Cutoff Yield

The five cumulative variance ratio functions $F_i{}^C(V_X)$ depending on the yields all showed sigmoidal forms (Figure 2), which fit the Richards' equation in a very highly significant way ($p < 0.01$) with a correlation coefficient $R^2$ close to 0.99 (Table 2).

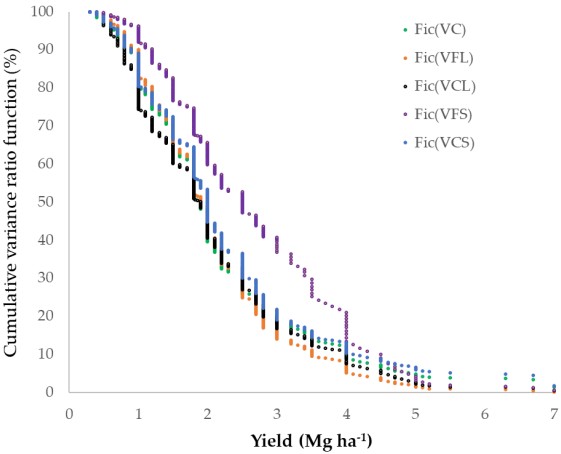

**Figure 2.** Relationship between wheat yield and cumulative variance ratio function.

The inflection points from these equations vary from 1.36 to 1.99 Mg ha$^{-1}$. The highest yield close to 2.0 Mg ha$^{-1}$ was used to separate the sub-population of the acceptable yield from those less acceptable. In the same context, several studies have pointed out that cereal production in Algeria during the period 2010–2017, is estimated on average at 4.12 million Mg. Compared to the decade 2000–2009, the production was estimated on average at 3.26 million Mg. During the period (2010–2017), cereal including durum wheat recorded an average of 2.04 Mg ha$^{-1}$ compared to the yield obtained during the 2000–2009 period, which did not exceed 1.4 Mg ha$^{-1}$ [30–32], which was far from

covering the growing demand. This production variability is directly and indirectly influenced by morphological, physiological, and environmental factors [33]. Specimens from fields with an acceptable yield ≥2.0 Mg ha$^{-1}$ of durum wheat represented 167 of the 350 specimens belonging to the entire study population.

*6.2. Step 2. Norms Derived from the Acceptable-Yield Subpopulation*

The total population of 350 specimens showed that 167 are above and 183 are below the yield threshold of 2.0 Mg ha$^{-1}$ in the first step; hence, the proportion of low yield specimens is 183 of 350 or 52%. If we assume that the sampled population is normal and its median value is close to its average value of the critical yield of 2.0 Mg ha$^{-1}$, the 50th percentile would be close to 52%. We deduce a critical chi-square value of 4.2 (Figure 3) corresponding to a maximum to qualify a sample in the sub-population with an acceptable durum wheat yield. This critical value not only identifies the appropriate texture to obtain an acceptable yield, but also the order in which the other textures could become limiting to the productivity of cereals in the semi-arid region of eastern Algeria. This new indication of critical chi-square could constitute an element of diagnosis and recommendation of soils suitable for this culture in this region. This integration of the textural components into a single global index (Equation (7)) could generate compositional standards specific to each agro-pedo-climatic system.

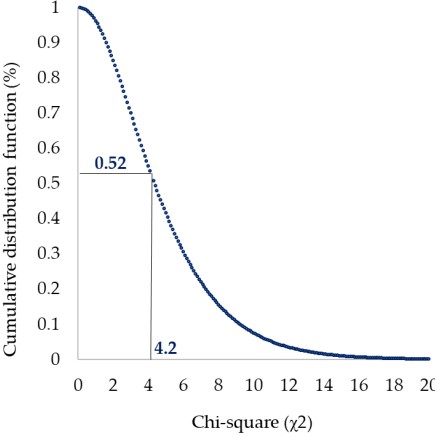

**Figure 3.** The chi-square cumulative distribution function with 5 df to obtain theoretical threshold. $r^2_{granulo}$ value of 4.2 for yield cutoff at 52% of low-yield sub population.

*6.3. Step 3. Validation of the Threshold Particle Size Imbalance Index*

The granulometric compositional standards are the means of the centered log values calculated on the 167 specimens of the population with an acceptable yield greater than or equal to 2.0 Mg ha$^{-1}$, and the standard deviations of these same centered log values, but over the entire sampled population of 350 specimens (Table 3).

**Table 3.** Norms for the acceptable-yield wheat subpopulation (2.0 Mg ha$^{-1}$), assuming a less acceptable-yield subpopulation proportion of 52% (texture norms are the means: V*$_C$, V*$_{FL}$, V*$_{CL}$, V*$_{FS}$, and V*$_{CS}$ and standard deviations: SD$_C$, SD$_{FL}$, SD$_{CL}$, SD$_{FS}$, and SD$_{CS}$).

|  | V*$_C$ | V*$_{FL}$ | V*$_{CL}$ | V*$_{FS}$ | V*$_{CS}$ |
|---|---|---|---|---|---|
| **Mean** | 0.316 | 0.143 | 0.047 | 0.229 | −0.736 |
| **SD** | 0.626 | 0.716 | 0.485 | 0.994 | 1.126 |

As expected, Figure 4 shows a downward trend in wheat yield as a function of the single granulometric index $r^2_{granulo}$. The binary partition of Cate-Nelson in this figure divides the sub-populations with acceptable yields from those with lower yields.

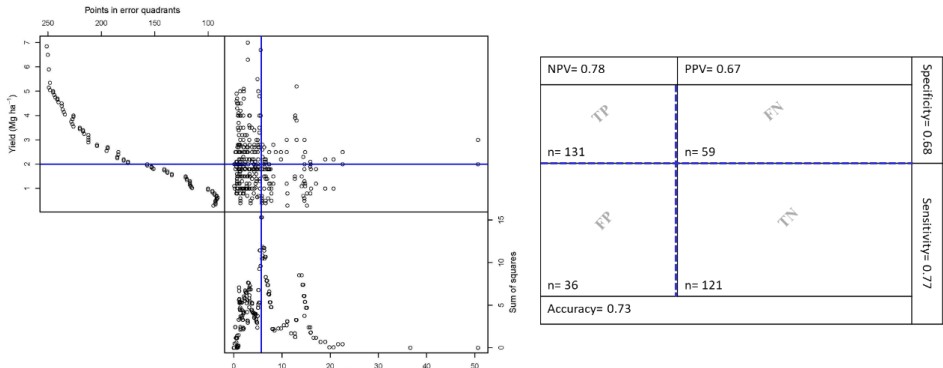

**Figure 4.** Relationship between particle size index Compositional ($r^2_{granulo}$) and wheat yield, and summary table (lower right) n: Number of points in the different quadrants; performance indicators of the partition model. Accuracy; NPV: negative predictive value; PPV: positive predictive value.

The critical value of the $r^2_{granulo}$ is 4.6, dividing between a balanced particle size composition from the unbalanced particle size composition. Based on this critical value of $r^2_{granulo}$, we can classify the semi-arid regional soils of eastern Algeria into two textural groups: (i) the balanced texture group, which granulometric $r^2_{granulo}$ is less than 4.6, normally having good physical fertility, leading to acceptable yield levels ≥2.0 Mg ha$^{-1}$ by the agricultural farmers of this region; and (ii) the group of unbalanced texture, in which $r^2_{granulo}$ is higher than 4.6, having a poor physical fertility, with a less acceptable yield (<2.0 Mg ha$^{-1}$). The high probabilities of acceptable yields lie in the true positive (TP) and false negative (FN) quadrants. The error quadrants (FP) correspond to acceptable yields despite belonging to an unbalanced texture class, while the TN quadrants correspond to a less acceptable yield, although it is in the balanced textural class. In the Figure 4, we obtained a high probability (73%) of a correct diagnosis for a balanced texture of the studied soils. This value is defined as the point's ratio in the TP and FN quadrants to the points total number in the four quadrants. The specificity [TP/(TP + FN)], which represents the probability of making the right decision (the right texture) compared to all observations with yield stability for the obtained model in Figure 4, was 68%. The sensitivity [TN/(TN + FP)] represents the probability of choosing the right texture compared to all the observations with a less acceptable yield (<2.0 Mg ha$^{-1}$). This sensitivity value is 77%, and it is the probability that lower yields will occur for soils that have an unbalanced texture. The positive predictive values (PPV) [TN/(TN + FN)] represent the probability of a positive yield response to a balanced texture when the critical value is less than 4.6. This PPV is 67% (Figure 4). The negative predictive values (NPV) [TP/(TP + FP)] are the probability that the wheat crop is not suitable for an unbalanced texture of a higher $r^2_{granulo}$ of 4.6. This NPV (78%) is high to consider the correct diagnosis of physical soil fertility in eastern Algeria (Figure 4). The PPV and NPV allow us to say agronomically that about 3 to 4 out of 5 cases of a specimen diagnosed with a balanced or unbalanced texture are in the high or low yield subpopulation, respectively. All values of the five probabilities (accuracy, specificity, sensitivity, PPV, and NPV) of the compositional texture diagnostic test are between 0 and 100%. A perfect diagnostic test has a probability of 100%, while a non-discriminatory test has a probability of less than 50%. In general, these five probabilities allow us to say that the precision of this type of diagnosis is considered sufficient to good, according to the scale established by Šimundić [34]. This author considers that the diagnostic accuracies tests are excellent, very good, good sufficient, bad, and not useful, and are in the following probability ranges, respectively: [90–100%], [80–90%], [70–80%], [60–70%], [50–60%], and [0–50%].

*6.4. Step 4. Particle Size Ranges*

We proceeded for the particle size ranges in the same way as Figure 4, with a binary partition in two groups to separate each of the textural indices ($I^2_C$, $I^2_{FL}$, $I^2_{CL}$, $I^2_{FS}$, and $I^2_{CS}$). These binary partitions gave a value of 0.7 for $I^2_C$, 1.9 for $I^2_{FL}$, 0.1 for $I^2_{CL}$, 0.2 for $I^2_{FS}$, and 1.7 for $I^2_{CS}$.

The sum of the squared critical granulometric indices was, therefore, 4.6 and it coincides with the $r^2_{granulo}$ obtained in step 3.

This is a cross-validation of the critical value $r^2_{granulo}$, defining the soil granulometric balance for cereal sustainability in eastern Algeria. Applying these critical indices to Equation (6), the critical $r^2_{granulo}$ gives:

$$r^2_{granulo} = 0.7 + 1.9 + 0.1 + 0.2 + 1.7+ = 4.6. \tag{7}$$

This same compositional approach applied to the same four steps on plant nutritional indices resulted in the same precision of the cross-validation obtained by Khiari, Parent, and Tremblay [21,22].

From these five critical squared indices, we deduce the symmetrical intervals of each of these granulometric indices: ±0.83 for $I_C$, ±1.73 for $I_{FL}$, ±0.31 for $I_{CL}$, ±0.44 for $I_{FS}$, and ±1.30 for $I_{CS}$ (Table 4).

The two fractions that seem to have the most influence on the textural imbalance are fine silt ($I_{FL}$) and coarse sand ($I_{CS}$) since their contribution is 41% (1.9/4.6) and 37% (1.7/4.6), respectively, to the critical value of the global imbalance index.

**Table 4.** Critical particle size indices and yield cutoff in the validation.

| Index | Critical Value | Critical Range | |
|---|---|---|---|
| | | **Lower Limit** | **Upper Limit** |
| $I_C^2$ | 0.7 | −0.83 | +0.83 |
| $I_{FL}^2$ | 1.9 | −1.37 | +1.37 |
| $I_{CL}^2$ | 0.1 | −0.31 | +0.31 |
| $I_{FS}^2$ | 0.2 | −0.44 | +0.44 |
| $I_{CS}^2$ | 1.7 | −1.30 | +1.30 |
| Sum $r^2_{granulo}$ | 4.6 | 0 | 4.2 |

Samples using the Cate-Nelson partitioning procedure.

The imbalance caused by the silt could be explained by its sensitivity to the problem of reduced porosity, air, and water circulation, thus causing a reduction in root elongation.

This phenomenon is commonly called the slaking crust, and it causes significant drops in wheat yields [14].

As for the second component of coarse sand, its contribution to the textural imbalance could be explained by its poor capacity for retaining water and nutrients. Under conditions where there is very little coarse sand, Wambeke [13], Prasad et al. [35], Ludwig and Asseng [36], Asseng et al. [37], Jalota et al. [38], Gu et al. [39], and Daryanto et al. [40] have observed very low wheat yields.

## 7. Conclusions

The soil quality diagnosis in Algeria is essential for efficient and sustainable agricultural production. The most essential parameter for making recommendations for cereal use in Algeria is the texture, since it greatly influences the production, and this is represented by five particle size fractions forming a closed composition simplex denoted $S^5$.

A compositional analysis consisting of making log-centered ratio transformations of these fractions, followed by a determination of the granulometric indices of each fraction, and at the end a globalization by a granulometric imbalance index noted $r^2_{granulo}$, led to acceptable relations with the wheat yield.

This compositional analysis took place in four complementary steps, ensuring a robustness of this granulometric diagnosis. From the first step, five cumulative variance ratio functions $F_i^c (V_X)$ allowed us to establish an acceptable yield level of 2.0 Mg ha$^{-1}$.

From the second, a theoretical critical chi-square granulometric index of the order of 4.2 is calculated to qualify a sample in the sub-population with an acceptable wheat yield.

From the third, a Cate-Nelson binary partition is established to deduce the critical granulometric index $r^2_{granulo}$, defining the soil granulometric balance for cereal sustainability in eastern Algeria.

From the last step, the critical $r^2_{granulo}$ gives five critical intervals of each granulometric component where two fractions, which have the most influence on the textural imbalance, are fine silt ($I_{FL}$) and coarse sand ($I_{CS}$).

**Author Contributions:** Drafting of the original project, methodology, and planning of the experiment, S.Z. and L.K.; Execution of field and laboratory work, S.Z., Modelling, J.G., S.Z. and L.K. Editing and revision, S.Z., L.K., S.A., J.G., G.K. and supervision, L.K., S.A. and F.K. All authors have read and agreed to the published version of the manuscript.

**Funding:** The financing of the field work was provided by the company TELL SARL SMID. The cost of training on the CND concept and modelling techniques was covered by the Natural Sciences and Engineering Research Council of Canada (NSERC) (No. RDCPJ528053-18).

**Acknowledgments:** The authors thank KERAGHEL A. the CEO of the TELL SARL SMID, as well as the farmers of the RéQuaBlé Sétif network, particularly MEHNANE S. and BOURAS M.S. for their help in the data collecting.

**Conflicts of Interest:** The authors declare no conflict of interest.

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
