# Peer review of "Particle Size Imbalance Index from Compositional Analysis to Evaluate Cereal Sustainability for Arid Soils in Eastern Algeria"

_agriculture, doi:10.3390/agriculture10070296_

Round 1

Reviewer 1 Report

Dear authors

The manuscript entitled “Development of global indicators of soil texture by compositional analysis within cereal agrosystems in eastern Algeria”, the authors showed the relationships between wheat crop yield and soil particle distribution. I have many concerns for the manuscript especially, the lack of background explanation.

The background is unclear. The content of the sentence has suddenly changed from L45, therefore, I can't keep up. I don't understand the subsequent explanations, hypotheses, results, and arguments because of the lack of enough explanation here.

The authors suggested that the particle distribution of the soil affects wheat yields. I didn’t understand how you got the hypothesis and why did you ignore the relationships between soil chemical properties and crop yields. Several studies indicated that soil chemical properties especially N, P and K strongly correlated with crop production. Why not consider it there?

In the result section, the PPV and NPV were 67% and 78%, respectively. It is not possible to determine whether the percentage is high or not. In addition, this is only a result of using obtained data and not a decision based on verification data. How versatile is it to put together a formula that is convenient for the current data? There is a lack of evaluation there.

In the title, what does "Global" in the title mean? This time, I only saw the soil of a certain area. So why can we say global? If you want to use “Global”, you must consider and compare several soils, crop species, and fields.

The unit needs to be an international unit. This is very basic: Qx is not a common unit.

Author Response

Thank you sir for your relevant comments. Regarding work, the article is a part of a whole doctoral work on the quality of agricultural soils with the title (development of quality indicators for cereal soils in some semi-arid regions of eastern Algeria). Certain soil properties could be good indicators if taken together. The choice of these indicators requires the selection of the properties most sensitive to changes in cultural practices. To constitute good indicators, the selected soil properties must be easy to measure, verifiable and well linked to land use and environmental transformation. In our research, particular soil properties were chosen as potential indicators according to different agricultural practices.
 Physical indicators of the soil: The apparent density, the water retention, the granulometry and as well as the structural stability were chosen as basic physical characteristics of the soil and to study the effects of different cultural practices (tillage, fertilization, precede crop and irrigation).
 Soil chemical indicators: Among the chemical indicators of soil quality, soil reaction (pH). Soil organic matter. The cation exchange capacity and the assimilable phosphorus.
 Soil biological indicators: Soil microorganisms have been shown to be potentially useful (early and sensitive) indicators of soil health, as they respond to soil management at time scales (months / years) relevant to the land management
So sir we have chosen for the article just one indicator among a set of indicators selected for the doctoral work.

Reviewer 2 Report

Broad Comments:

  1. As with the study sites, it is not clear where the yield was measured. From each of the 350 plots?
  2. How does the authors justify that these textural classifications are specific to the cereal systems? How is that achieved?
  3. Overall application of the results is vague in the manuscript. How will these results be used? In conjunction with fertility management or predicting yield thresholds?

Specific Comments

There are several language and style editing required to make the manuscript flow readable . I have highlighted few. Please make changes appropriately.

Line 35: consider removing “including”

Line 37 : Line 45: Line 209 :  I did not understand the contextual meaning of “ cultural vocation” in this manuscript.

Line 77: Consider using “ volumetric mass”

Author Response

  1.  We took the yield of each plot of these 350 plots (350 yield obtained)
  2. For each plot studied, soil samples were taken from a depth of 25 cm (worked horizon), where they subjected to a series of physico-chemical analyzes, including particle size analysis, to determine the textural class. of each plot. in order to then determine the relationship between the different textural fractions and the wheat yield. The results obtained show that, the plots, which have a large percentage of fine silt and coarse sand, are the same plots, which record an acceptable yield. These results are statistically confirmed where r2granulo gives five critical limits; the two fine silt and coarse sand fractions influence the yield more compared to the other fractions (Clay, coarse loam and fine sand) and, are the same characteristics of the soils of the semi-arid regions of eastern Algeria. Other works are launched, to give more validity to these results by comparing with other cultures.
  3.  The work is original in Algeria despite, the age of the subject of soil quality in the world. For this reason the results are somewhat vague since the main objective was to create a platform of indicators of the qualities of agricultural soils in Algeria (physical, chemical and biological indicators) to diagnose the condition of agricultural land towards more sustainable, we started with cereal soils because it is the most popular crop in Algeria. According to these results, we have proposed other research projects to give more precision to these results.

 Specific Comments

Line 35:  the word is deleted

Line 37 : Cultural vocation means the occupation of the plots studied

Line 77:The work, sir, is a thesis for obtaining a doctorate in agricultural science. The subject is to develop physical, chemical and biological quality indicators for cereal soils. Among the physical indicators we have chosen the grain size, the apparent density and the structural stability. Just for the article sir we presented only one indicator

Reviewer 3 Report

Dear authors,

I think that the manuscript need a better proofreading (some sentences are not very clear). The Reference list contains several mistakes and typos and the citation style is not always right (especially for the books).

Some information about Compositional Data/analysis are needed in the introduction.

The statistical Analysis section need to be greatly developed (e.g. Richard’s equations and curve, Cate-Nelson binary partition…). I also think that it should be nice to do some yield predictions based on these indicators by analysing a calibration and a validation datasets separately instead of using all samples for developing the indicators. For example by leaving alone (as a validation set) 50 or 100 samples and predict their yields with the indicators. These yields could then be compared to the real ones.

I have included also several minor comments below.

Title:

You should probably add semi-arid region in the title since "Global Indicators" is a little misleading.

Abstract:

L13-14: please reformulate that sentence. Also, in that sentence you are speaking of validating those Indicators. It should be nice to have real validation.

Introduction:

I believe you but some references are not possible to find anywhere. Maybe you could add some more available ones.

L37: ion exchange capacity.

L37: “cultural vocation” is a wrong denomination. Please reformulate.

L43: Besides ≫ Moreover

L44: examine the textural…

L45: Reformulate “soil cultural vocation”

L60: What do you mean by the soils are carbonates? Reformulate.

L61: calcids and cambids soils. Which taxonomy? Pleas complete.

L63: practiced farming systems ?

L67-68: reformulate

L71: The particle size distribution

L74: then a sample was made. It sounds strange. Can you explain?

L83: Table 1: Wrong title (Basic statistics). Also, I can find anywhere in the text mention of that Table !

L84 and everywhere: please change all “loem” for “loam”

L87: The Statistical Analysis section need to be greatly updated.

L99: expect in Table 1 caption, all these abbreviations are not defined anywhere (C, FL,…) also could you include somewhere the textural composition of these classes if it is not an international classification.

L100: Something wrong with that sentence.

In equation (2), it is difficult to read and understand with all those “x”.

L110: I could be wrong but SD signification (standard definition) is nowhere to be found.

Results and Discussion:

In Figure 2, the yellow is hard to detect.

Table 2. PI means inflection point? What are Fc(PI) and Yield (PI).

Table 2, L132: it is not clear what you mean for B and the yields (lower…)

L147: an histogram with the yields would be nice.

Figure 4: the signification of TP , FN,… is missing in the caption and also the % for PPV.

L225: reformulate In abundant…

References:

There are many errors, mistakes and typos here. Some Journals are with abbreviations, some with the full name etc.

Author Response

Some information about Compositional Data/analysis are needed in the introduction.

The statistical Analysis section need to be greatly developed (e.g. Richard’s equations and curve, Cate-Nelson binary partition…). I also think that it should be nice to do some yield predictions based on these indicators by analysing a calibration and a validation datasets separately instead of using all samples for developing the indicators. For example by leaving alone (as a validation set) 50 or 100 samples and predict their yields with the indicators. These yields could then be compared to the real ones.

I have included also several minor comments below.

Title:

it's good the word is added

Abstract:

L13-14: We rephrased the sentence

Introduction:

I don't understand what to tell you

L37: The texture of the soil controls the phenomenon of exchange of certain ions between the Argilo-humic complex and the solution of the soil

L37:  It’s done

L43:  it’s done

L44: examine the textural = study textural

L45: Reformulate “soil cultural vocation” = for the choice of crops

L60:  It’s done

Carbonate soils are soils that lie above limestone rocks. The soil carbonates are calcite (calcium carbonate) and dolomite (double carbonate of calcium and magnesium). this is the type that characterizes the majority of soils in semi-arid regions in Algeria

L61: according to the USDA American classification , the cambids are the Aridisols with the least degree of soil development. These soils have a cambic horizon within 100 cm of the soil surface.  Calcids soil are the Aridisols with calcium carbonate that was in the parent materials or was added as dust, or both. Precipitation is insufficient to leach or even move the carbonates to great depths. The upper boundary of the calcic or petrocalcic horizon is normally within 50 cm of the soil surface. If the soils are irrigated and cultivated, micronutrient deficiencies are normal. these two types are the soils that characterize the semi-arid areas of eastern Algeria

L63: In the areas studied, the cropping systems are: monoculture; crop rotation (vegetable cultivation / wheat, Barley / wheat and fallow / wheat)

L67-68: it’s done

L71: it is the distribution of different soil fractions (Clay, loam and sand)

L74: I don’t understand

L83: it’s done

L84: it’s done

L87: I don’t understand

L99: It’s just a table that represents the characteristics of the soils studied to give a general vision on the nature of the plots

L100: It’s done

In equation (2), it's corrected

L110: I could be wrong but SD signification (standard definition) is nowhere to be found. The SD is standard deviation, it has been calculated most often according to an equation derived by Eq. 4

Results and Discussion:

In Figure 2,  It’s corrected

Table 2. Fc (PI): this is the inflection point of the cumulative functions of each particle size fraction with the inflection point of the Yield PI yields, where we level from this point, we can separate between the most acceptable yield and least acceptable for the plots studied

Table 2, L132: B is the threshold obtained to separate the three levels of yield: low yield, acceptable yield and high yield

L147: an histogram with the yields would be nice.

Figure 4: It’s corrected

L225: it’s corrected

References:

it’s corrected

Reviewer 4 Report

Interesting study, based on large research material, very good use of statistics to compile research result. Some naming errors. The weak point is the list of references - full names of journals in one place, abbreviations in another.

Specific comment:

  • whether the fields set aside in Algeria are left in natural succession or are they managed in a controlled way, e.g. non-agricultural crops, fertile plants
  • line 30 and all papers - why quintals and not tones?
  • line 80 – assimilable? better to use „available”
  • line 84 - incorrect spelling: Loem? – loam; balk? – bulk
  • line 85 – incorrect spelling: Netrogen? – Nitrogen; CaCo3? -- CaCO3
  • line 135 – Loem? – Loam
  • line 251 - Semcheddine, N. (2015) but line 252 -  Bencherif S., 2000; year of publication written in brackets or without? - this remark applies to the entire list of references
  • a similar remark regarding the entire list of publications applies to the name of journals, e.g line 267: Soil Science Society of America Journal  but line 271: Soil Sci. Soc. Am. J. ;
  • the names of journals always write with a capital letter – line 264/265: Bulgarian journal of agricultural science? - Bulgarian Journal of Agricultural Science
  • conversely, we write the titles of works with a lowercase letter – line 281: Compaction of Coarse …..Compaction of coarse …
  • with some entries in the list of references there are DOI numbers - should they be in all

Author Response

Comments and Suggestions for Authors

Interesting study, based on large research material, very good use of statistics to compile research result. Some naming errors. The weak point is the list of references - full names of journals in one place, abbreviations in another. it's good we have corrected the refferances

Specific comment:

  • whether the fields set aside in Algeria are left in natural succession or are they managed in a controlled way, e.g. non-agricultural crops, fertile plants
  • line 30 and all papers - why quintals and not tones?   Quintaux since it is the frequent unit in Algeria to calculate the yield
  • line 80 – assimilable? better to use „available”: we used the concept of assimilable phosphorus to estimate the share of total phosphorus available to plants. It’s done
  • line 84 - bulk it’s done
  • line 85 – it’s done
  • line 135 – it’s done
  • line 251 -  it's good we have corrected the refferances
  • a similar remark regarding the entire list of publications applies to the name of journals, e.g line 267: it's good we have corrected the refferances
  • the names of journals always write with a capital letter – line 264/265: it's good we have corrected the refferances
  • conversely, we write the titles of works with a lowercase letter – line 281: It's done

Round 2

Reviewer 1 Report

There are a number of points that do not reflect the reviewer's comments. Especially the title, the introduction, and the reference style in the sentence. If they do not comply, you should politely answer the reasons. In the absence of this, this response is disingenuous and I must reject the manuscript.

When replying to a comment, you must describe where and how you corrected it. Your comment alone doesn't quite get there.

It's fine to comment here on the Introduction, but the response needs to be reflected in the paper.

In particular, you mustexplain why you focused on particle size and not chemical properties in your paper. In your reply, you said it was because it was part of my PhD thesis, but that's not the reason on this paper. With that in mind, please add to Introduction the reason for the focus on physicality again.

Also, you're misunderstanding the meaning of “Global”. The other reviewer also says “Global” is a misleading, so delete this one. Why don't you obey?

I don’t know what is “Qx”?. Since the paper is read by people all over the world, it is necessary to use SI units. Therefore, it needs to be converted to “t” or “kg”. This point is also pointed out by other reviewers, but why not follow it?

The way the reference is quoted in the sentence is odd. Why don't you read the journal's regulations? Rewrite with reference to other papers.

Author Response

View attached files

Reviewer 3 Report

Please add a few sentences (with references) in the introduction concerning compositional analysis in similar fields of research.

In the Statistical Analysis section, you need to develop (give more information) about the cut-off values, Richard's equation, and the Cate-Nelson partion.

L13: ...improve the knowledge of...

L62: ... according to USDA classification [39]/

L86: Table 1. Soil parameters of the study area

Table 2: add the explanation you gave me to the manuscript.

Author Response

View attached files

Round 3

Reviewer 1 Report

 The manuscript was well improved:

L63: What is r2granulo? Please describe the details.

Author Response

This unique comment "L63: What is r2granulo? Please describe the details." is corrected as follows

It is therefore hypothesized that the compositional transformation of the five particle sizes (clay, fine loam, coarse loam, fine sand, and coarse sand) into a global index, noted r2granulo [20-22], is relevant if this global index can be related to cereal yields. This r2granulo index could then help the scoring process to assess soil health and cereal suitability.